# Advances in Microscopic Studies of Tendinopathy: Literature Review and Current Trends, with Special Reference to Neovascularization Process

**DOI:** 10.3390/jcm11061572

**Published:** 2022-03-13

**Authors:** Łukasz Jaworski, Maria Zabrzyńska, Anna Klimaszewska-Wiśniewska, Wioletta Zielińska, Dariusz Grzanka, Maciej Gagat

**Affiliations:** 1Department of Histology and Embryology, Faculty of Medicine, Collegium Medicum in Bydgoszcz, Nicolaus Copernicus University in Toruń, 85-092 Bydgoszcz, Poland; lukaszmjaworski@gmail.com (Ł.J.); maria.zabrzynska@gmail.com (M.Z.); w.zielinska@cm.umk.pl (W.Z.); 2Private Center of Surgery, 87-100 Toruń, Poland; 3Department of Clinical Pathomorphology, Faculty of Medicine, Collegium Medicum in Bydgoszcz, Nicolaus Copernicus University in Toruń, 85-094 Bydgoszcz, Poland; anna.klimaszewska@cm.umk.pl (A.K.-W.); d_gr@me.com (D.G.)

**Keywords:** tendon, tendinopathy, microscopy, neovascularization, Bonar

## Abstract

Tendinopathy is a process of chaotic extracellular matrix remodeling followed by increased secretion of enzymes and mediators of inflammation. The histopathological assessment of tendinous tissue is crucial to formulate the diagnosis and establish the severity of tendon degeneration. Nevertheless, the microscopic analysis of tendinous tissue features is often challenging. In this review, we aimed to compare the most popular scales used in tendon pathology assessment and reevaluate the role of the neovascularization process. The following scores were evaluated: the Bonar score, the Movin score, the Astrom and Rausing Score, and the Soslowsky score. Moreover, the role of neovascularization in tendon degeneration was reassessed. The Bonar system is the most commonly used in tendon pathology. According to the literature, hematoxylin and eosin with additional Alcian Blue staining seems to provide satisfactory results. Furthermore, two observers experienced in musculoskeletal pathology are sufficient for tendinopathy microscopic evaluation. The control, due to similar and typical alterations in tendinous tissue, is not necessary. Neovascularization plays an ambiguous role in tendon disorders. The neovascularization process is crucial in the tendon healing process. On the other hand, it is also an important component of the degeneration of tendinous tissue when the regeneration is incomplete and insufficient. The microscopic analysis of tendinous tissue features is often challenging. The assessment of tendinous tissue using the Bonar system is the most universal. The neovascularization variable in tendinopathy scoring systems should be reconsidered due to discrepancies in studies.

## 1. Tendon Histology and Pathology

Tendons are structures necessary to distribute the force generated by muscles [1,2,3]. They are subjected to extraordinary loads, which may lead to the development of pathology. Tendons are designed to contribute to human body movements, to stabilize joints, and to absorb the kinetic energy [3]. The structure of tendon is firm and fibro-elastic in texture, with a brilliant white color [4]. There are various types of tendons, such as flat, cylindrical, fan, and ribbon shaped [3]. Franchi et al. emphasized that short and thick tendons are responsible for transmitting high torques or resistive forces. On the contrary, long and thin tendons are responsible for soft and delicate movements [3]. The adaptive properties of tendinous tissue result from the highly organized structure, which can be described as a morphology resembling a synthetic climbing rope [5]. This special construction of tendons into subsequent subunits ensures a more uniform spread of loads and decreases the possibility of damage [5,6].

In the structure of the tendon, two elements can be distinguished: cells and extracellular matrix. The extracellular matrix acts as a scaffolding for cells, vessels, and nerves and is responsible for the strength of tendon [4,7]. It contains collagen molecules together with some non-collagenous substances, e.g., elastin, glycosaminoglycans, proteoglycans, glycoproteins, and water [8,9]. Type I collagen accounts for approximately 60–90% of the tendon dry mass. However, some other collagen types can also be found in the tendon structure, which are mainly III and V types [4,5]. Type I collagen is responsible for tendon tensile forces. Synergistically with type III, it forms heterotypic fibrils. Type III collagen is involved in tendon healing and forms cross-links. On the other hand, type V collagen forms a core for type I collagen fibrils and controls the lateral growth of the tendon [8].

Moreover, the collagen fibrils are surrounded by loose connective tissue, called endotenon. It complements the tendon structure and functions as energy storage. In turn, every collagen fascicle is surrounded by another connective tissue sheath: peritenon. These connective tissue elements provide access to blood and lymphatic vessels and innervation. The whole tendon is covered by epitenon, which is an additional connective tissue barrier [10].

The most frequent non-fibrous proteins in tendons are proteoglycans, which constitute 1–5% of a tendon dry mass [6,9,11,12]. Their main components are core proteins connected with polysaccharide chains, known as glycosaminoglycans side chains. Due to the possibility of binding large amounts of water, they guarantee strong hydration of the tendon. As a result, 55–70% of its mass is water. The most abundant proteoglycans are small leucine-rich types, of which decorin is the most widespread. Additionally, some other proteins, e.g., fibromodulin, biglycan, lumican, and keratocan, occur in smaller amounts. They lubricate and separate the elements of the tendon and participate in fibrillogenesis and matrix assembly. Other extracellular matrix proteins are glycoproteins. Among them, the most common is the cartilage oligomeric matrix protein (COMP) [6,13,14,15]. Glycoproteins control the cell-to-cell interactions as well as adhesion and transduce mechanical stimuli to the cytoplasm in the mechanotransduction process [6]. During the mechanotransduction process, tendons collect information about the load and signals from the local environment. As a result, the metabolism of tenocytes and the influence on the local mediators can be easily regulated [5]. Tendon disorders can be divided into acute and chronic [1,5]. Various factors may increase the risk of tendinous tissue pathology. Among them are sex, age, smoking, chronic diseases (e.g., type 2 diabetes, thyroid diseases, rheumatologic diseases, inherited diseases, mellitus, obesity), nutrition, occupation, hyperthermia, fluoroquinolone antibiotics, corticosteroids, joint instability, bone impingements, and tendon overloading during exercise, sport, and other physical activity [4,5,11,16,17,18]. Both acute and chronic injuries may completely or partially break tendon continuity. However, acute injuries have a higher probability of complete regeneration [19,20]. In turn, chronic injuries, which arise from overload and lead to tissue degeneration, are known as tendinopathies. They cause mechanical dysfunction of the tendon [21]. Tendinopathy is often associated with physical exercises and represents approximately 30–50% of all injuries connected with sport [1,22]. Injuries or repeated strain may lead to pain and swelling, which usually cause difficulties in movement [23,24]. Joseph et al. noted that tendinopathy is characterized by impaired healing of the extracellular matrix, with a limited amount of inflammatory cells, collagen degeneration, and increased proteoglycans and type III collagen levels [2,25,26]. Collagen fibers lose their hierarchical structure and become more irregular and spacious. Moreover, some pathological alterations of cells, non-collagenous ECM, nerves, and the vascular bed appear [17,21,27,28,29]. There is no clear link between clinics and histopathology in tendinopathy.

Similarly, the ultrastructure of tenocytes in tendinopathy has revealed a few characteristic changes. They become randomly scattered around cells, with disrupted localization and morphology of the nuclei [7,30]. The transformation of the shape of tenocytes into a more oval shape could be evidence of cartilage metaplasia and adaptation. It is thought that the cytoplasm and cellular shape variations reflect the gradual chondroid transformation [7,29]. Furthermore, some authors have noted the presence of apoptotic-like features in tenocytes, such as chromatin alterations and nuclei fragmentation [4,31].

Regarding the ECM, a replacement of collagen fibrils by non-collagenous ECM, with a subsequent decrease in the number of crimps, has been observed [6]. Tendinopathy is a process of a chaotic extracellular matrix remodeling and the increased secretion of enzymes, proteins, and mediators of inflammation, such as metalloproteinases, TNF-α, IL-1β, IL-6, IL-10, VEGF, TGF-β, and PGE2 [2,25]. There is a group of tendons especially prone to injuries due to their specific biology and localization [1,5,18,24,32,33,34,35]. Usually, these tendons are exposed to increased forces with various vectors of action, e.g., rotator cuff tendons, long head of the biceps tendon, Achilles tendon, posterior tibialis tendon, patellar tendon, gluteal tendons, and tibialis anterior tendon [26,35,36,37,38,39]. Moreover, these tendons contain hypovascular or avascular areas, so-called watershed regions, additionally predisposing them to pathology [2].

In this review, we aimed to compare the most popular scales used in tendon pathology assessment and reevaluate the role of the neovascularization process in tendinous tissue degeneration.

## 2. Classification of Histopathology and Current Trends

The histopathological assessment of tendinous tissue is crucial to formulate a diagnosis and establish the severity of tendon degeneration [29,40]. Nevertheless, microscopic analysis of the tendinous tissue features is often challenging and difficult. In this review, we present a few of the most popular scales used in tendon pathology assessment: the Bonar score, the Movin score, the Astrom and Rausing Score, and the Soslowsky score (Table 1).

Assessment using the Movin and Bonar systems is the most commonly used in tendon pathology [28,29]. The variables in the Movin scale are the fiber structure, fiber arrangement, rounding of the nuclei, regional variations in cellularity, increased vascularity, decreased collagen stainability, hyalinization, and glycosaminoglycans content. Each parameter is scored with a four-point scoring system. The final score is between 0 (healthy tendon) and 24 (severely degenerated tendon). Tissue can be classified as slightly abnormal when the final score is between 1 and 8; moderately abnormal when the final score reaches values between 9 and 16; markedly abnormal when the final score is between 17 and 24. On the other hand, the variables in the Bonar scale are tenocyte morphology alterations, ground substance accumulation, extent of neovascularization, and collagen bundle architecture. Each variable is scored with a four-point scale, where 0 indicates healthy tissue and 3 indicates severely degenerated tissue. The total score ranges from 0 (normal tendon) to 12 (the most severe abnormality).

The less popular scales are the Astrom and Rausing Score and Soslowsky score [40,41,42]. The Astrom and Rausing Score is also a semiquantitative score to evaluate tendinopathy. It includes five parameters scored with a four-point scale. The final score varies between 0 (normal tendon) and 20 (the most severe degeneration). The Soslowsky score, used to investigate rotor cuff diseases, considers five histologic features of tendinosis, scored with a four-point scale. The final score reaches values between 0 (normal tendon) and 16 (the most severe degeneration).

These scoring systems introduce similar variables, but they do not determine some of the pivotal issues. These include specimen preparation, staining methods, additional immunohistochemical reactions, the number of investigators, the experience of investigators, the area of investigation, and a certain magnification. These features have not yet been well-established, and they are usually selected randomly by authors [27,28,37,43,44,45,46,47].

Regarding the staining methods, the H&E is the gold standard, but some authors support it with Alcian Blue and Masson Trichrome staining to visualize the ECM alterations [21,29,48,49,50]. Moreover, some of the authors have included in the evaluation of tendinous tissue IHC techniques to assess neovascularization, using CD31 and CD34 [44,50] type I collagen [51], or the apoptosis process [52].

The number of microscopic investigators in various papers differs from one to three observers. However, in the majority of articles, the number of investigators was not specified [27,28,37,43,51]. In our opinion, at least two observers experienced in MSK pathology are necessary to avoid bias. The inter-observer reliability in the Bonar system was evaluated by Fearon et al. The authors revealed that after a few modifications, the revised Bonar score was characterized by satisfactory inter-tester reliability; r2 = 0.71 [28].

Furthermore, authors usually randomly select the evaluated area of the slide. In some papers, the most severely degenerated area was chosen, while in other cases the entire slide was evaluated [29,44,47,49]. Regarding the modifications of the scoring systems, the four-variable Bonar scale was most commonly applied. Some of the authors introduced a fifth variable, counting the number of tenocytes [28,37,51,52,53]. Additionally, in some studies, a control group was included. Alterations in the course of tendinopathy are usually similar in all tendon structures. The characteristic changes include disrupted collagen architecture, tenocyte morphological alterations, neovascularization process, and expansion of the ground substance. Thus, the control group should not be obligatory, especially when experienced musculoskeletal observers assess the specimen.

## 3. The Issue of Neovascularization

The neovascularization process is typical for osteoarthritis, retinopathy, inflammation, tumors, and tendon disorders [18]. In healthy tendons, vascularization is modest, with a small number of capillaries localized between bundles of collagen in the ECM. These capillaries mainly arise from the musculotendinous junction, osteotendinous junction, and connective tissue sheath [54]. Tendons are metabolically active structures, and like other tissues, they require a blood supply. However, the vascular perfusion is relatively weak compared to other types of connective tissue [16,54]. Moreover, tendons usually contain specific hypovascular regions, which heal poorly and are especially prone to degeneration [55]. On the other hand, tendinopathy manifests microscopically as a degenerative process. It is also characterized by the expansion of newly formed capillaries, followed by chaotic production of ECM components [17,44,56].

Recent studies have revealed that neovascularization in tendon disorders has a mythological status and does not necessarily agree with clinics [55,57]. The Achilles tendon is the most often injured tendon in humans. Its tendinopathy usually occurs with an abundant neovascularization process [58]. Hypothetically, concomitant nerve ending ingrowth is responsible for pain in this disorder. Neurovascular ingrowth in clinically painful areas of tendons has been described in a few studies, which revealed a positive correlation of the pain level and the density of capillary vessels [16,59,60]. Numerous authors presented US-guided techniques to decrease the level of neovascularization and inject the sclerosants in the most vascularized areas of the tendon [58]. However, some studies clearly showed that there is no connection between neoangiogenesis and pain [44,61,62]. Neovascularization was also found in Achilles tendons among asymptomatic athletes [63].

Some authors observed in a tendinous tissue a surprising phenomenon: cigarette smoking inhibits the neovascularization process [17,64]. Although the level of neovascularization in smoking individuals showed greater resemblance to healthy tissue and underestimated the microscopic evaluation score, it was not consistent with the clinical outcome. Nevertheless, nicotine increases the neovascularization process in macular degeneration, which is associated with obstetric complications in pregnancy or spontaneous miscarriage [65,66,67]. The issue of smoking shows the importance of the neovascularization process in tendinopathy. However, the microscopic evaluation of the pathological role of the new vessel formation is not simple and may not be linked with clinics. Regarding the pathology and microscopic assessment of tendinous tissue, Fearon et al. revealed that the four-variable Bonar score may be biased by several factors [28]. The authors suggested that the complete lack of vascularity in the obtained pathological tendons, as well as the excessive expansion of new capillaries, should be graded as 3 points (extreme pathology). Moreover, the area of the neoangiogenesis investigation is also a subject of dispute. Some authors assess the most pathological area of tendinous tissue, while others evaluate the entire specimen [28,44,48,49]. Furthermore, some also supported their studies with immunohistochemical methods, with the use of CD31, CD34, factor VIII-related antigen, CD105, or smooth muscle actin [44,68,69]. Tendinous tissue after the injury undergoes a regeneration process, which consists of three main phases. In the formation phase, an intensive neoangiogenesis is usually observed [54] (Figure 1).

The process is specifically regulated by the local microenvironment components, such as mediators and ECM [62]. However, the exact levels of these mediators are still unknown. The ECM regulates the topology and elongation of vessels, using the proteases to pave the way for new capillaries [70]. Some authors observed that angiogenesis is significantly reduced in the areas of the tendinous tissue, with the increased production and aggregation of non-collagenous ECM [17,71] (Figure 2 and Figure 3).

Koehler et al., using a 3D angiogenesis model, demonstrated that glycosaminoglycan accumulation impaired the biological activity of VEGF [72]. Furthermore, Cheng et al. revealed that non-collagenous components of the ECM inhibit VEGF receptor signaling [73]. VEGF levels are elevated after the inflammatory phase, during the formation and remodeling phases [5]. Its main role is the stimulation of the angiogenesis process [74]. The neovascularization process is crucial in the tendon healing process. On the other hand, it is also an important component of the degeneration of tendinous tissue, when the regeneration is incomplete and insufficient.

## 4. Conclusions

The microscopic analysis of the tendinous tissue features is often challenging and difficult. The assessment using the Bonar system is the most commonly used in tendon pathology. According to the literature, H&E staining with additional Alcian Blue seems to provide sufficient material for the observer. Moreover, two investigators experienced in musculoskeletal pathology ensure a satisfying microscopic evaluation of tendinopathy. The control, due to similar and typical alterations in tendinous tissue, is not necessary. Neovascularization plays an ambiguous role in tendon disorders. Thus, this uncertain variable should be reconsidered and updated in tendinopathy scoring systems.

## Figures and Tables

**Figure 1 jcm-11-01572-f001:**
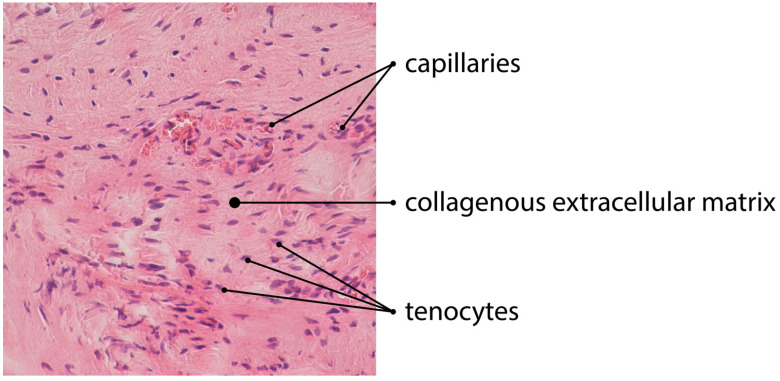
H&E staining of tendinous tissue showing angiogenesis (magnification of the objective: 20×).

**Figure 2 jcm-11-01572-f002:**
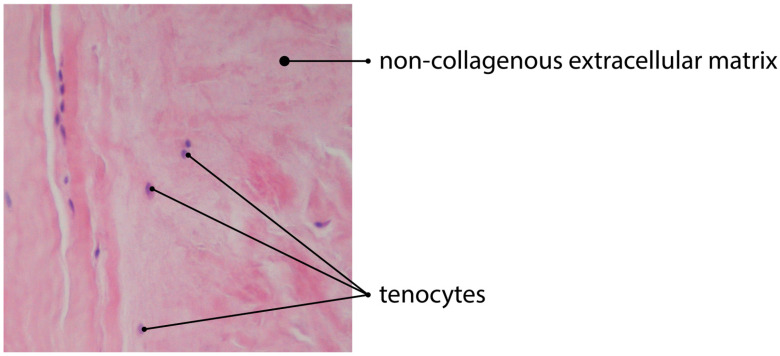
H&E staining of tendinous tissue showing impaired angiogenesis due to deposition of non-collagenous ECM (magnification of the objective: 20×).

**Figure 3 jcm-11-01572-f003:**
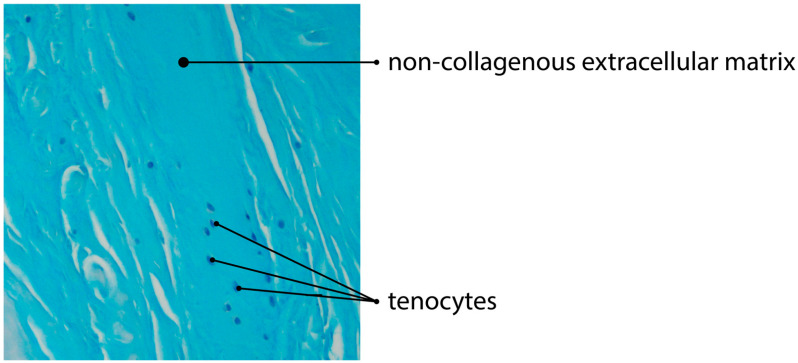
Alcian blue staining of tendinous tissue showing impaired angiogenesis due to deposition of non-collagenous ECM (magnification of the objective: 20×).

**Table 1 jcm-11-01572-t001:** The most popular scoring systems in tendon pathology assessment.

Scale	Items	Points
Movin	Fiber structure	0–3
	Fiber arrangement	0–3
	Rounding of the nuclei	0–3
	Regional variations in cellularity	0–3
	Increased vascularity	0–3
	Decreased collagen stainability	0–3
	Hyalinization	0–3
	Glycosaminoglycans	0–3
Bonar	Tenocyte morphology	0–3
	Ground substance	0–3
	Vascularity	0–3
	Collagen	0–3
Astrom and Rausing	Fiber alignment	0–3
	Fiber structure	0–3
	Morphology of tenocyte nuclei	0–3
	Variations in cell density	0–3
	Variations in cell density	0–3
Soslowsky	Cellularity	0–3
	Fibroblastic changes	0–3
	Collagen fiber orientation	0–3
	Disruption	0–3

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
