# Peer review of "Advances in Microscopic Studies of Tendinopathy: Literature Review and Current Trends, with Special Reference to Neovascularization Process"

_jcm, 2022, doi:10.3390/jcm11061572_

Round 1

Reviewer 1 Report

- In my opinion the overall aim of the study could be clearer. It mixes aspects of a narrative review with a systematic approach and eventually does not really add extensive new information to the field.

- According to recent consensus statements tendinopathy is a clinical condition and the histopathology is not always related to the symptoms. Even though histological analyses was/is important for understanding the pathology the common belief nowadays seems to be that treating the symptoms is much more important than trying to reverse the tissue. I therefor suggest to include a statement where the correlation between pain and histopathology is clarified.

- Lines 180-193 (and in abstract): Avoid personal opinions. This seems not to be a systematic review and therefore one should be careful with drawing "personal" conclusions.

- Figures: Add some marks into the figures to clarify which structures (and where) are seen

Author Response

Thank you for the opportunity to improve and resubmit our manuscript. Your suggestions have been immensely helpful. We appreciate the clarification of the concerns about the paper.

1. In my opinion the overall aim of the study could be clearer. It mixes aspects of a narrative review with a systematic approach and eventually does not really add extensive new information to the field.There is no clearly defined aim of the work.

Thank you for the comment, we set clear aims of the study.

2. According to recent consensus statements tendinopathy is a clinical condition and the histopathology is not always related to the symptoms. Even though histological analyses was/is important for understanding the pathology the common belief nowadays seems to be that treating the symptoms is much more important than trying to reverse the tissue. I therefor suggest to include a statement where the correlation between pain and histopathology is clarified.

Line 109 – the sentence was added.

3. Lines 180-193 (and in abstract): Avoid personal opinions. This seems not to be a systematic review and therefore one should be careful with drawing "personal" conclusions.

The abstract was rewritten.

4. Add some marks into the figures to clarify which structures (and where) are seen

The important structures on micrographs were labeled.

Reviewer 2 Report

This paper reviewed the current scoring system used for tendinopathy. The authors, further, described the concerns of those scoring systems with relation to the neovacularization process. This is well written and I have no problem for the publication. 

Table 1

Soslowsky score which is less popular came on the top in this table. Please change the order of the score as described in the paper. 

Author Response

Thank you for the opportunity to improve and resubmit our manuscript. Your suggestions have been immensely helpful. We appreciate the clarification of the concerns about the paper.

1. Table 1 Soslowsky score which is less popular came on the top in this table. Please change the order of the score as described in the paper. 

Thank you for the comment. We changed the order of Soslowsky score.